# Immobilization of the Peroxygenase from *Agrocybe aegerita*. The Effect of the Immobilization pH on the Features of an Ionically Exchanged Dimeric Peroxygenase

Diego Carballares [1,†], Roberto Morellon-Sterling [1,†], Xiaomin Xu [2], Frank Hollmann [2,*] and Roberto Fernandez-Lafuente [1,3,*]

1 Departamento de Biocatálisis, ICP-CSIC, Campus UAM-CSIC Cantoblanco, 28049 Madrid, Spain; diego.carballares@csic.es (D.C.); r.m.sterling@csic.es (R.M.-S.)

2 Department of Biotechnology, Delft University of Technology, 2629HZ Delft, The Netherlands; X.Xu-2@tudelft.nl

3 Center of Excellence in Bionanoscience Research, External Scientific Advisory Academic, King Abdulaziz University, Jeddah 21589, Saudi Arabia

* Correspondence: f.hollmann@tudelft.nl (F.H.); rfl@icp.csic.es (R.F.-L.)

† Both authors have evenly contributed to this paper.

**Abstract:** This paper outlines the immobilization of the recombinant dimeric unspecific peroxygenase from *Agrocybe aegerita* (r*Aae*UPO). The enzyme was quite stable (remaining unaltered its activity after 35 h at 47 °C and pH 7.0). Phosphate destabilized the enzyme, while glycerol stabilized it. The enzyme was not immobilized on glyoxyl-agarose supports, while it was immobilized albeit in inactive form on vinyl-sulfone-activated supports. r*Aae*UPO immobilization on glutaraldehyde pre-activated supports gave almost quantitative immobilization yield and retained some activity, but the biocatalyst was very unstable. Its immobilization via anion exchange on PEI supports also produced good immobilization yields, but the r*Aae*UPO stability dropped. However, using aminated agarose, the enzyme retained stability and activity. The stability of the immobilized enzyme strongly depended on the immobilization pH, being much less stable when r*Aae*UPO was adsorbed at pH 9.0 than when it was immobilized at pH 7.0 or pH 5.0 (residual activity was almost 0 for the former and 80% for the other preparations), presenting stability very similar to that of the free enzyme. This is a very clear example of how the immobilization pH greatly affects the final biocatalyst performance.

**Keywords:** ionic exchange; enzyme immobilization; enzyme stability; effect of immobilization medium on enzyme immobilized stability

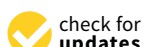



## 1. Introduction

The modern use of enzymes as catalysts in the organic chemistry industry is receiving growing interest due to their high activity under mild conditions (that makes them suitable from an environmental point of view) along with their high specificity and selectivity (that can save purification and protection/deprotection steps) [1–4]. Nevertheless, enzymes have been optimized by nature to fulfill physiological functions and are not necessarily well suited for the needs of industrial biotransformation [5]. To be efficient catalysts for chemical transformations, enzymes need to be active and stable under the reaction conditions (which may differ significantly from the physiological ones) while at the same time converting non-physiological substrates [6–8]. Recovery and reuse of enzymes after the reaction may be also a problem, as they are water-soluble catalysts [9–11].

These factors have considerably delayed the industrial implementation of enzymes; fortunately, there are many tools to improve the enzyme features that have experienced significant advancements in recent years, offering new solutions to solve natural enzyme limitations [12]. Advances in enzyme modelling along with site-directed mutagenesis can

improve targeted features of enzymes in a controlled way [13] or the design of plurienzymes [14]. Metagenomics permits the use of existing biological diversity, even of uncultivable or no longer existing organisms [15–17], while directed evolution can mimic natural evolution but also focus on selected enzyme features and can be achieved rapidly [18–20]. The understanding of the enzyme chemistry has led to the development of strategies for fully directed chemical modification [13,21–24].

In this context, enzyme immobilization was initially proposed as a solution for the problem of enzyme recovery and reuse [14,25,26]. However, in recent years it has become a powerful tool to improve other enzyme features (stability, activity, selectivity, specificity, inhibitory issues, etc.) [6,27–30] and even to purify enzymes [31].

Enzyme rigidity may be increased via multipoint covalent attachment; this usually results in an increase of the enzyme stability [30]. In the case of multimeric enzymes, the inactivation may be initialized by the dissociation of enzyme subunits or by the promotion of erroneous subunit assembly [32]. The first inactivation cause may be prevented by involving the immobilization all enzyme subunits [32], though the second inactivation cause may even be increased if the immobilization generates some tension on the enzyme assembly [33].

Both multi-subunit and multipoint enzyme immobilization may become a very complex goal. First, it is necessary to select a proper immobilization system: Protocol, support and reactive groups on the support must be adequate [34]. The final support surface should be as inert as possible to prevent undesired enzyme–support interactions; ideally, the only groups able to interact with the enzyme should be those introduced by the researcher [35]. Agarose beads are a support that fulfils these requirements [36]. Moreover, the reactive groups on the support surface need to be properly selected to ensure a high probability of intense multipoint attachment [34]. They should be stable, without steric hindrances for the reaction with the groups in the enzyme, and able to react with groups located on the enzyme surface (usually primary amino groups) [34]. Among those described in literature, glyoxyl [37], vinyl sulfone [38] and glutaraldehyde [39]-activated supports are usually recognized as suitable for yielding an intense multipoint attachment.

Glyoxyl-agarose is involved in the immobilization of non-protonated primary amino groups (from the terminal amino group of the enzyme and the lateral chain of Lys), giving reversible imine bonds [37]. It requires the enzyme immobilization at alkaline pH value, as the first immobilization needs to be multi-punctual or the enzyme molecule is not incorporated to the support [40]. Moreover, the final step of the immobilization process is a reduction with sodium borohydride to convert the reversible imine bonds in irreversible secondary amino bonds and the remaining aldehyde groups in chemically inert and hydrophilic hydroxyl groups [37]. Both factors may become a problem if the enzyme is unstable at alkaline pH values or it is sensible to reduction. These problems may be partially solved using some stabilizing additives, such as polyols (e.g., glycerol) [41–44], or by using alternative reducing agents that do not affect the enzyme [45].

The vinyl sulfone group is able to react with the lateral chains of several amino acids, including thiol, imidazole, phenol and primary amine groups, giving very stable bonds [38,46,47]. Glutaraldehyde mainly reacts with the primary amino groups of the enzyme, but it establishes stable enzyme–support linkages through its capacity to form stable cycles [39]. Glutaraldehyde-activated supports are among the most versatile enzyme immobilization matrices [48–50].

The enzyme features also play a very important role in the possibility of obtaining an intense multipoint covalent attachment [34]. The groups able to react with the support should be abundant on its surface, and the presence of polysaccharide chains can seriously hinder the prospects of an intense multipoint covalent attachment by generating steric hindrances to the enzyme–support reaction [51]. For example, enzymes can be only immobilized in a glyoxyl support if there is an area in the protein surface with several Lys groups available to simultaneously react with the support [40]. The immobilization rate

on these supports depends in an exponential way on this intrinsic enzyme's structural features [52].

An alternative to the multipoint covalent immobilization is the enzyme immobilization by physical interactions with the support, which used to be a multi-punctual process [53]. In the case of lipases, their physical immobilization on hydrophobic supports via interfacial activation may be considered a very suitable immobilization protocol, as it permits their one step immobilization, purification, stabilization and hyper-activation [54]. However, for other enzymes, the immobilization via ion exchange is the most utilized physical immobilization strategy [53]. This immobilization protocol requires the establishment of several ionic enzyme–support interactions to fix the enzyme to the support [55]. The immobilization on supports coated with poly-ethylenimine (PEI) can lead to a strong immobilization of the enzyme in a polymeric bed; the enzyme may be stabilized by the generation of a hydrophilic environment around the enzyme (mainly against the action of hydrophobic molecules such as oxygen or organic solvents) [56]. The mobility of the polymeric bed permits its interaction with a large area of the enzyme surface, providing stronger immobilization than using small groups located in the support surface. The immobilization of enzymes via ion exchange can stabilize mainly multimeric enzymes if all enzyme subunits are involved in the immobilization, and if subunit dissociation is the first step of the enzyme inactivation, by preventing it [32]. However, it may be expected that the enzyme conformational rigidity will be hardly improved by immobilization using physical adsorption, as the energy of each individual enzyme–support bond is very low. As the surface of the support remains "active" during operation, this kind of immobilization may even have negative effects on enzyme stability, as incorrect structures of the enzyme induced by the experimental conditions may be stabilized [57,58]. It has been shown that the enzyme immobilization conditions (e.g., pH, ionic strength) on ionic exchangers may alter the final biocatalyst properties [50,59–61]. One of the advantages of this immobilization strategy is the possibility of releasing the inactivated enzyme after its inactivation during operation and reuse of the support for new enzyme immobilization cycles [53].

In this contribution, we investigated the immobilization of the peroxygenase from *Agrocybe aegerita* [62], particularly its recombinant, evolved variant (r*Aae*UPO) [63,64]. r*Aae*UPO is a member of the heme-thiolate (protoporphyrin IX containing ferrous ion also known as Heme B group) peroxygenase family [65,66]. Peroxygenases share the prosthetic Heme group with the well-known P450 monooxygenases [65,67–69]. Both enzyme classes utilize a highly oxidized oxyferryl species (so-called compound I, Cpd I) as an oxygen transfer agent for catalysis (see Scheme 1). P450 monooxygenases generate this group via reductive activation of molecular oxygen necessitating electron transfer from NAD(P)H via a complex electron transport chain. Peroxygenases generate Cpd I directly from $H_2O_2$, thereby circumventing the complex electron transport chain and simplifying their practical application.

Peroxygenases have been used as catalysts for selective hydroxylation, [70–82] epoxidation [83–86] and sulfoxidation reactions [87,88]. Engineered enzymes with tailored substrate specificity have been reported [64,80,81,89–91], and issues related to the oxidative inactivation in the presence of $H_2O_2$ have been solved as various in situ $H_2O_2$ generation systems are now available [92].

Immobilization studies with peroxygenases, however, are relatively scarce. r*Aae*UPO is a highly glycosylated, dimeric enzyme (see Figure 1 and Figure S1) [93,94]. To date, few studies have been undertaken to immobilize the enzyme [95–97]. Covalent immobilization on epoxide supports (IB-COV-1 from ChiralVision) was successful but yielded enzyme preparations with dramatically decreased catalytic activity [85]. More recently, a His-tagged variant has been immobilized on EziG carriers with respectable activity recoveries between 53% and 78% [98].

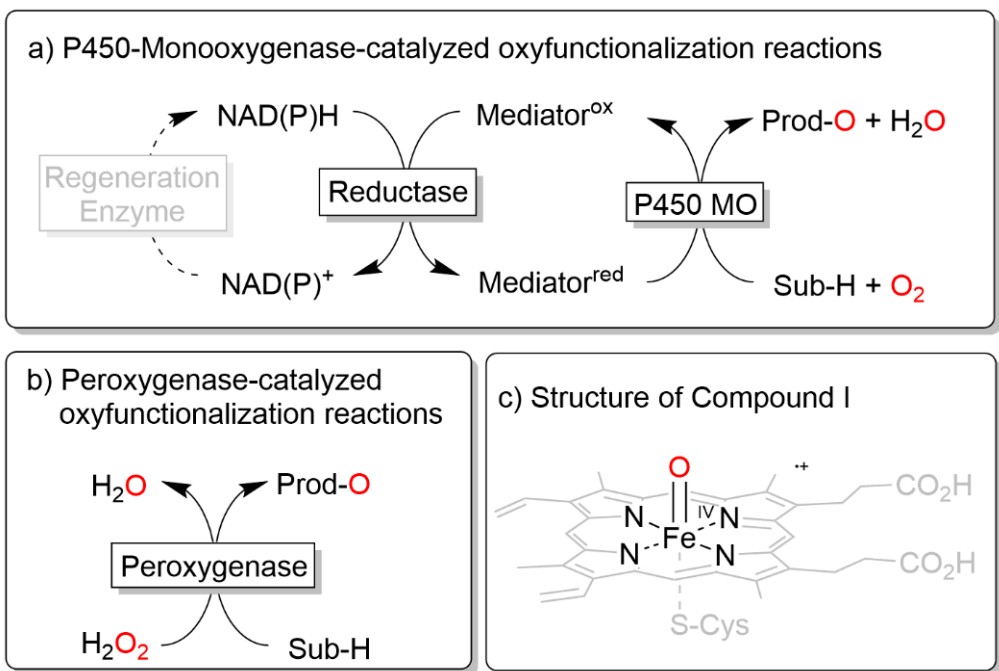

**Scheme 1.** Comparison of P450 monooxygenases (**a**) and peroxygenases (**b**) as catalysts for selective oxyfunctionalization reactions. Both enzyme classes utilize compound I (Cpd I, (**c**)) as an oxygen transfer agent.

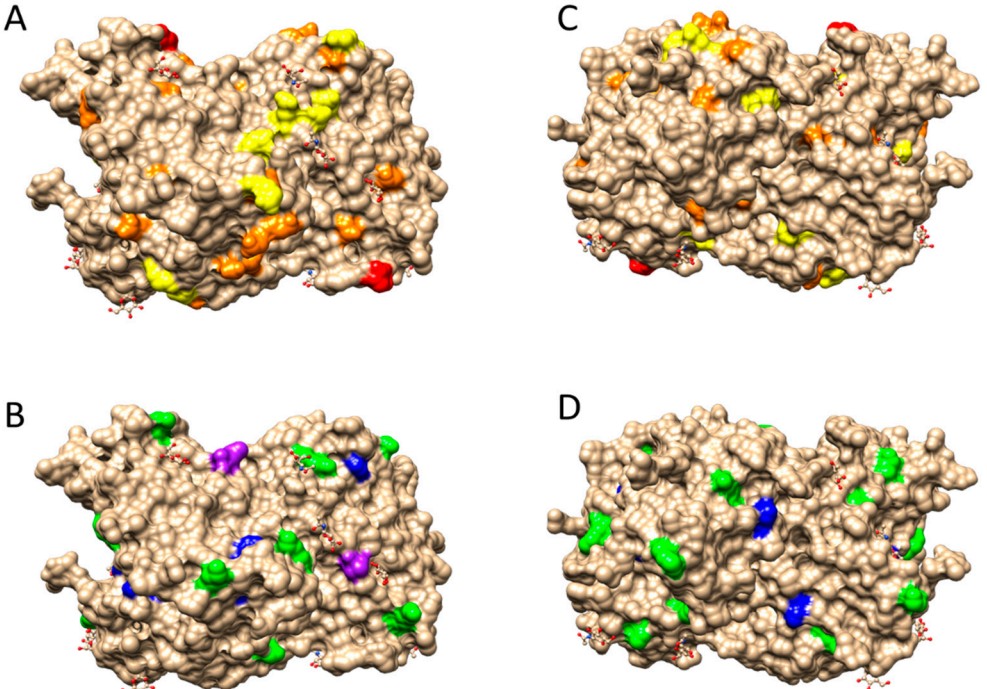

**Figure 1.** Structure of r*Aae*UPO. The figure shows a surface diagram of r*Aae*UPO showing some amino acids (green: lysine; blue: histidine; orange: aspartic acid; yellow: glutamic acid; red: C-terminal residue; purple: N-terminal residue) and the polysaccharide chains (balls and sticks) of side 1 (**A**,**B**) and the opposite side (**C**,**D**) of the enzyme. Sides were chosen by random.

Thus, in this contribution, the range of conditions under which the enzyme may be handled was studied as well as the effect of some additives on its stability. r*Aae*UPO immobilization on different supports via covalent immobilization (using glyoxyl, vinyl sulfone

or glutaraldehyde-activated agarose) or physical exchange (on PEI or monoaminoethyl-N-aminoethyl (MANAE)-activated agarose beads) was assayed. The expressed activity and stability of the biocatalysts were evaluated.

## 2. Results and Discussion

### 2.1. Handling of rAaeUPO

Before starting the immobilization trials, the range of conditions where the enzyme was stable and can be handled must be determined. The enzyme activity was determined under the conditions reported in literature (pH 4.4) [63,64]. At pH 7.0, the observed activity versus ABTS was only 10% of this maximum activity. The enzyme was very stable at pH 7.0, with not decrease in its activity even at 47 °C for 35 h. At pH 7.0, a strong dependence of the enzyme stability on the buffer was found. At 50 °C, the enzyme lost 50% of the activity after 3 h when 50 mM sodium phosphate was employed as buffer (Figure 2A), while if Tris or HEPES buffers were employed, the enzyme retained 100% of the initial activity. At 57 °C, the enzyme was inactivated at a similar rate using HEPES or Tris (Figure 2B). This suggested that phosphate presented a negative effect for the enzyme stability. This negative effect of phosphate anions has been described for other enzymes even in an immobilized form [99,100].

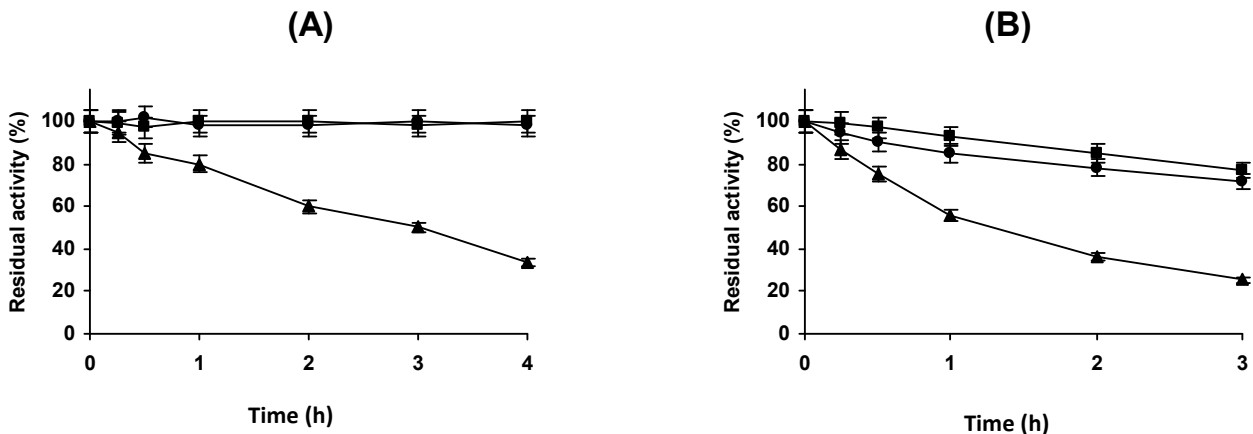

**Figure 2.** Effect of the buffer nature on the stability of free r*Aae*UPO. Inactivation was performed at pH 7.0 and (**A**) 50 or (**B**) 57 °C. Other specifications can be found in the Methods section. Full squares: 50 mM HEPES; full circles: 50 mM Tris; full triangles: 50 mM sodium phosphate.

This enzyme is dimeric; thus, we investigated whether the enzyme stability depends on the enzyme concentrations in both phosphate and Tris buffers (Figure 3). In both buffers, the enzyme concentration did not present a relevant impact on the enzyme stability. This suggested that reversible dissociation was not the first step of the enzyme inactivation at pH 7.0 and that the negative effect of phosphate anions on enzyme stability does not favor this dissociation [32]. Considering that the Heme B group is deeply buried in the enzyme structure, it was most unlikely that its release could be the first step of the enzyme inactivation. Therefore, enzyme inactivation seems to be initialized by a distortion of the tridimensional structure or an induction of an erroneous enzyme assembly.

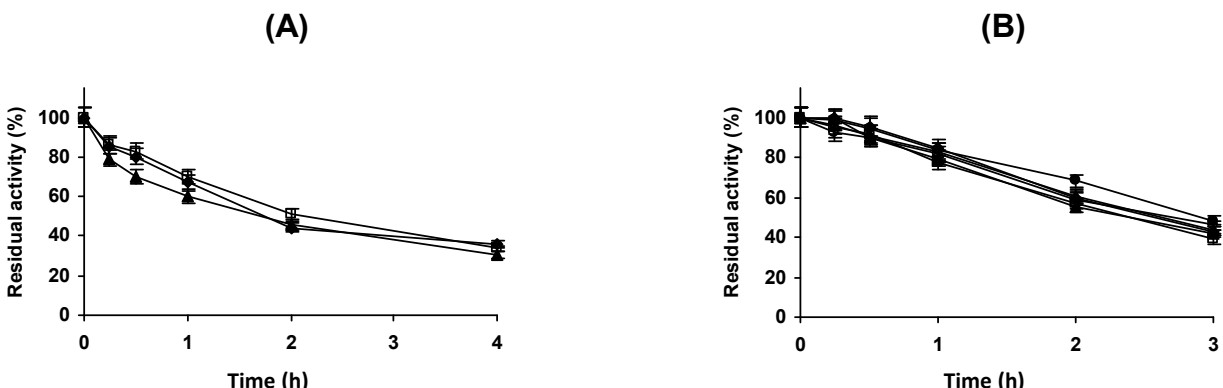

**Figure 3.** Effect of the concentration of free r*Aae*UPO stability. The experiments were performed at pH 7 (**A**) 50 mM sodium phosphate at 57.0 or (**B**) 50 mM Tris at 57.5 °C. Other specifications can be found in the Methods section. Full circles: dilution 1:250; full triangles: dilution 1:1000; empty squares: dilution 1:2000; full rhombus: dilution 1:4000; empty triangles: dilution 1:8000; empty circles: dilution 1:16,000.

At pH 10.0 (pH necessary to immobilize the enzyme on glyoxyl-agarose beads) [40] and 25 °C, the enzyme was rapidly inactivated (residual activity was only 5% after 4 h), preventing immobilization of the enzyme under these conditions (Figure 4). The addition of glycerol to the medium significantly stabilized the enzyme, and this was more significant when the glycerol concentration increased (the enzyme retained over 60% of the initial activity after 4 h in 40% glycerol) (Figure 4). This enzyme stabilizing effect of glycerol has been recently examined, and although it is not universal, it is a quite general phenomenon [44]. At 4 °C and in 40% glycerol, the enzyme maintained around 75% of its activity after 4 h (Figure 4).

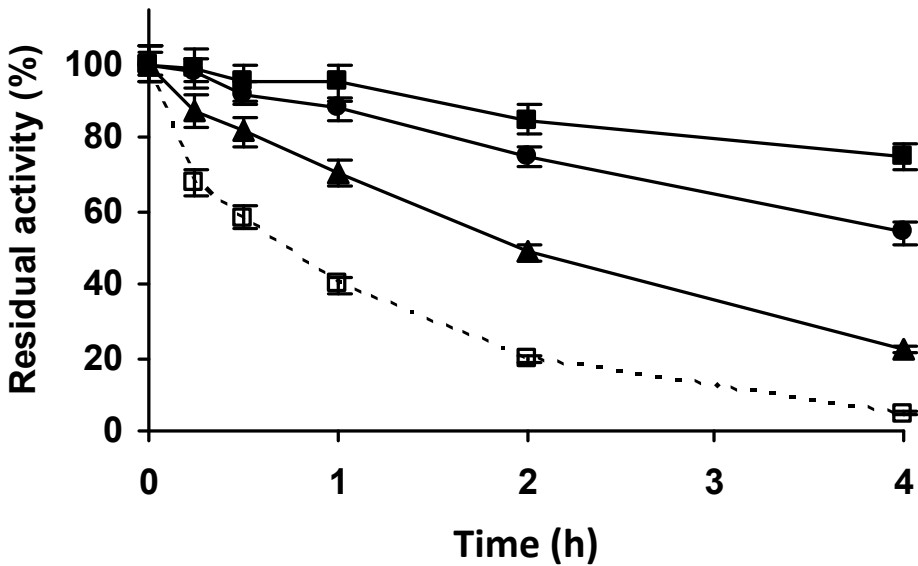

**Figure 4.** Inactivation courses of r*Aae*UPO at pH 10.0 under different conditions. The experiments were performed in 50 mM sodium carbonate. Other specifications can be found in the Methods section. Empty squares with dotted line: no additive at 25 °C; full triangles: 20 % glycerol at 25 °C; full circles: 40% glycerol at 25 °C; full squares: 40% glycerol at 4 °C.

We therefore found valid conditions to handle the enzyme at neutral and alkaline pH values, both necessary for the immobilization strategies that we intended to apply to this enzyme.

### 2.2. Covalent Immobilization of rAaeUPO

r*Aae*UPO was not immobilized in glyoxyl-agarose at pH 10.05 (Figure S3) (immobilization yield was almost 0). Looking to the structure of the enzyme (Figure 1), it can be seen that although the enzyme is rich in Lys residues areas, the polysaccharide chains may hinder the simultaneous interaction of several primary amino groups of the enzyme with the support, thus avoiding the enzyme immobilization in glyoxyl-agarose. The considerable dilution of the enzyme solution (almost 1000 folds) discards the possibility of some aminated compounds on the enzyme solution that could prevent the enzyme immobilization [101].

The enzyme was not significantly immobilized on supports activated with DVS at pH 5.0 and 7.0 (immobilization yield under 20%) (Figure 5A,B). Moreover, the small immobilized fraction decreased its expressed activity after immobilization by more than 50%, giving very poor final activity. At pH 9.0 (Figure 5C), a higher immobilization yield (over 90%) was observed, but the enzyme became almost fully inactivated after immobilization (expressed activity was under 10%). This enzyme inactivation upon immobilization was only partially reduced by adding 40% glycerol (expressed activity was 40%) (Figure 5D), but this additive also reduced the immobilization yield (to 50%). After 24 h, the expressed activity decreased to less than 5%, and if the biocatalysts were subjected to a blocking step (with ethylenediamine or aspartic acid), a necessary step to obtain a chemically inert support surface [38], the enzyme expressed activity fell below the detection limit.

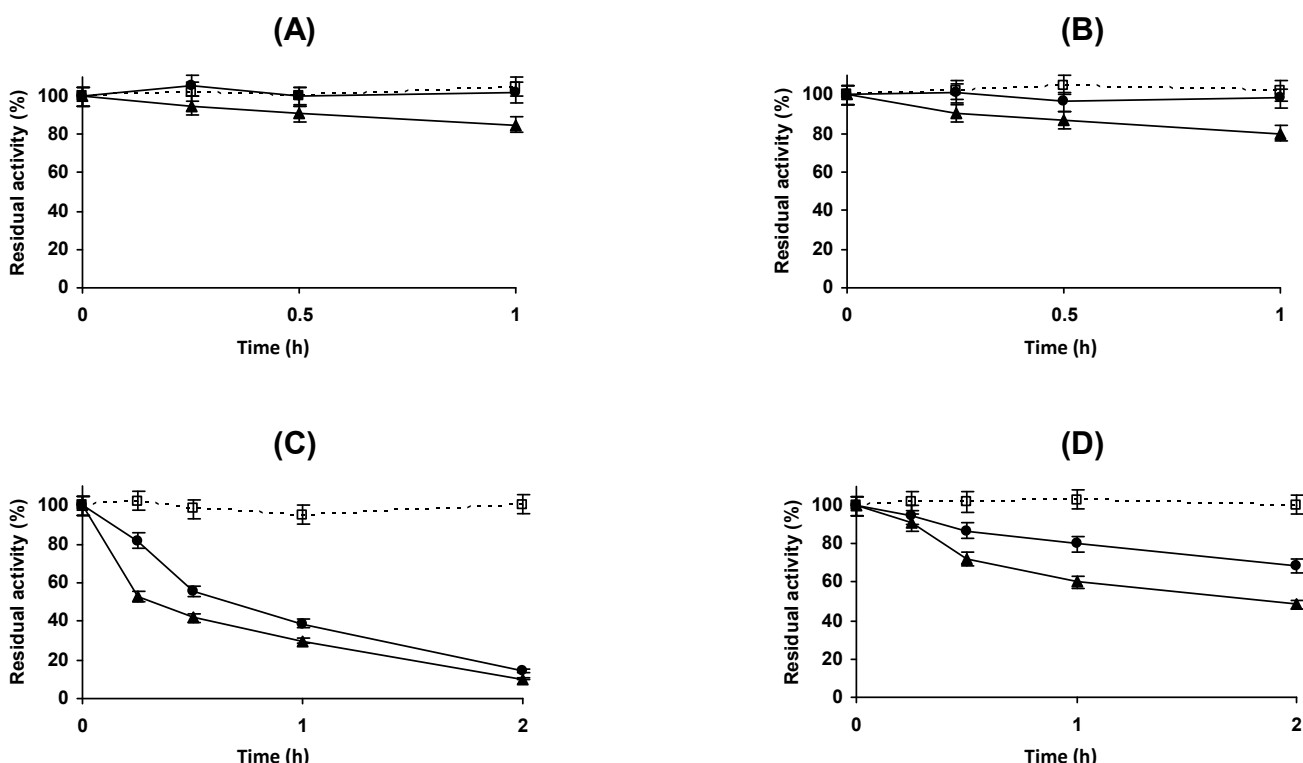

**Figure 5.** Effect of the immobilization conditions of r*Aae*UPO on agarose activated with vinyl sulfone support. The experiments were performed at 25 °C in (**A**) 50 mM sodium acetate pH 5.0, (**B**) 50 mM sodium phosphate pH 7.0, (**C**) 50 mM sodium carbonate pH 9.0 and (**D**) 50 mM sodium carbonate pH 9.0 with 40% glycerol. Other specifications can be found in the Methods section. Full circles: suspension; full triangles: supernatant; empty squares with dotted line: reference.

Thus, both covalent immobilization techniques were discarded for this enzyme.

Next, the enzyme was immobilized in pre-activated amino-glutaraldehyde supports at pH 5.0 and 7.0 (Figure 6). As the first step of the immobilization of enzymes in this support is the physical ion exchange of the enzyme on the support [48,102], this process was quite rapid, and the immobilization yield reached 100%. However, a significant percentage of activity was lost at both pH values, with expressed activities under 25% after 24 h of

incubation. Immobilization of the enzyme at pH 9.0 on this support resulted in an almost instantaneous enzyme inactivation. This decrease in enzyme activity could be explained by a certain enzyme–support multipoint covalent attachment. This should result in more rigid enzyme structure, and the decrease in activity could be compensated if this is correlated with an increase in enzyme stability [30,102,103]. However, as shown in Figure 7, the enzyme immobilized on this support was even less stable than the free enzyme. This decrease in the enzyme stability after immobilization on the support could be explained by the presence of hydrophobic and ionic groups near the enzyme surface that could stabilize some incorrect enzyme structures [35]. Another possibility is the production of some distortion of the assembly between both enzyme subunits caused by the multipoint covalent attachment, which could drive the higher tension of the multimeric structure and lower enzyme stability [33]. In any case, this immobilization strategy does not appear to be adequate to prepare an immobilized biocatalyst of this enzyme either.

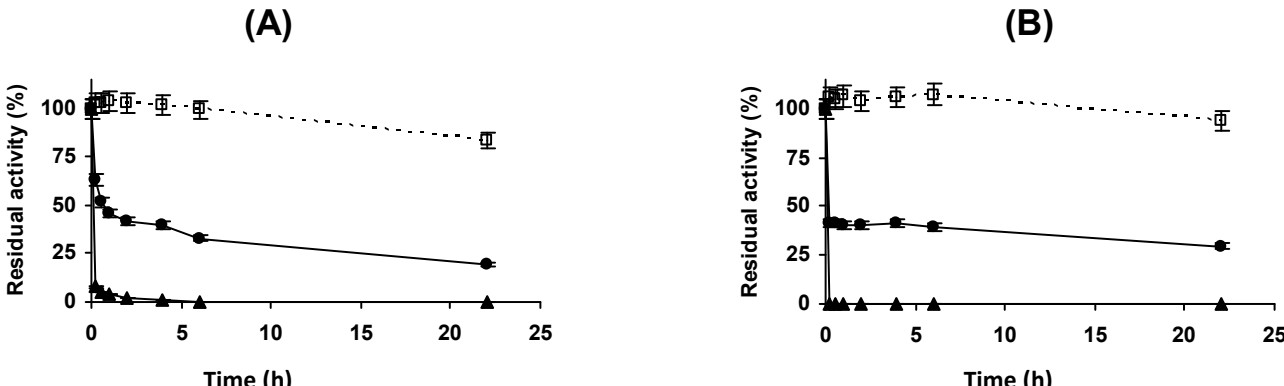

**Figure 6.** Effect of the pH on the r*Aae*UPO immobilization on amino-glutaraldehyde support. The immobilization was performed at 25 °C and (**A**) pH 5.0, (**B**) pH 7.0. Other details can be found in the Methods section. Full circles: suspension; full triangles: supernatant; empty squares with dotted line: reference.

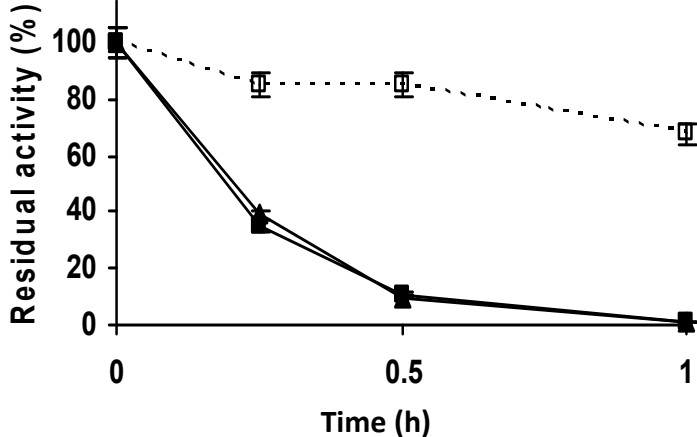

**Figure 7.** Thermal inactivation of r*Aae*UPO immobilized on amino-glutaraldehyde support compared to the free enzyme. The inactivation was performed in 10 mM Tris at pH 7.0 and 57 °C. Other specifications can be found in the Methods section. Full triangles: biocatalyst immobilized at pH 5.0; full squares: biocatalyst immobilized at pH 7.0; empty squares with dotted line: free enzyme.

### 2.3. Immobilization of rAaeUPO Via Ion Exchange

After the failure of the covalent immobilization protocols assayed above to produce a very active and stable immobilized biocatalyst using some covalent immobilization protocols, we assayed the r*Aae*UPO immobilization on MANAE-, PEI-10,000- and PEI-25,000-activated agarose at pH 7.0. The enzyme exposes various anionic groups in its

structure (see Figure 1) to permit the enzyme anion exchange on these cationic supports. Figure 8 shows the immobilization course in these supports. In all cases, immobilization is very rapid (immobilization yield was 100% after only 15 min). However, while in MANAE-activated agarose the activity is fairly preserved after immobilization (around 80%), and in PEI coated supports, a significant decrease of the enzyme activity is observed. After 48 h, these preparations did not maintain any activity even if maintained in the refrigerator at 6 °C just after immobilization. However, after that time, the MANAE biocatalyst maintained its activity intact even at 25 °C. This negative result observed when immobilizing the enzyme in PEI coated agarose could be explained if the flexible polymer was able to accede to internal pockets of the protein, which can be relevant for the enzyme activity or stability or even accelerate erroneous subunit assembly [56]. The amino groups attached to the surface of MANAE agarose should be unable to penetrate these internal pockets and would not produce this negative effect on enzyme stability, and the enzyme activity would be maintained. As shown in Figure 9, this biocatalyst, when inactivated at 57 °C and pH 7.0, has a similar stability to that of the free enzyme, or even marginally higher.

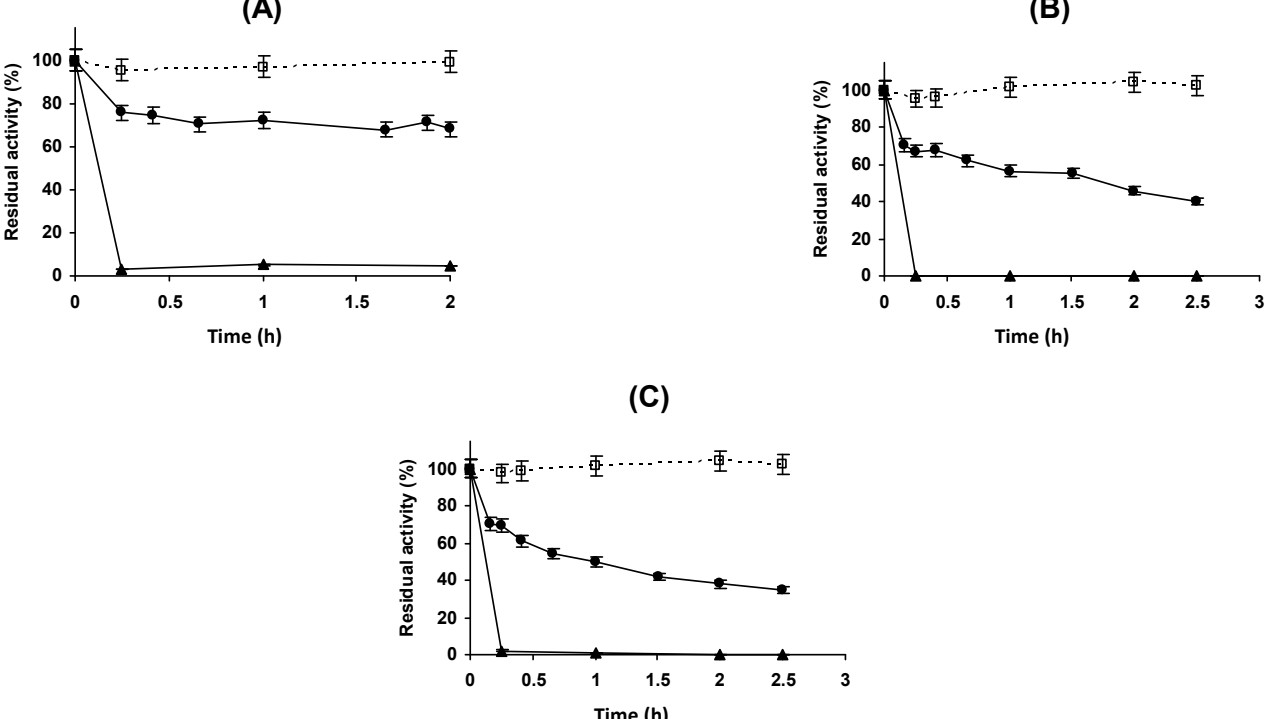

**Figure 8.** Immobilization courses of r*Aae*UPO at 25 °C and pH 7.0 on (**A**) MANAE agarose support, (**B**) PEI (10,000) agarose and (**C**) PEI (25,000) agarose. Other specifications can be found in the Methods section. Full circles: suspension; full triangles: supernatant; empty squares with dotted line: reference.

Considering the good properties of this biocatalyst compared to the covalent preparations, we tried to improve its performance using two different strategies.

First, the immobilization on MANAE agarose was also performed at pH 5.0 and 9.0 (Figure 10) and compared to that at pH 7.0, as it has been shown that the immobilization pH may greatly alter the final properties of the enzymes immobilized via anion exchange [48,61,102]. Compared to the immobilization of the enzyme at pH 7.0, the main difference is a higher expressed activity when immobilizing the enzyme at pH 5.0 (over 95%), while at pH 9.0 the expressed activity is around 80%, similar to the results obtained at pH 7 (Figure 8). However, immobilization yield was 90% for the immobilization performed at pH 5.0 and almost 100% if the immobilization was performed at pH 9. At pH 5.0, the cationic nature of the support is reinforced, while the anionic nature of the enzyme is de-

creased (the theoretic isoelectric point of the enzyme from the enzyme primary sequence is 5.325 (http://isoelectric.org/ (accessed on 04/04/2021))), and perhaps a small proportion of the enzyme molecules, which are highly glycosylated, cannot immobilize on MANAE supports under this less favorable pH value.

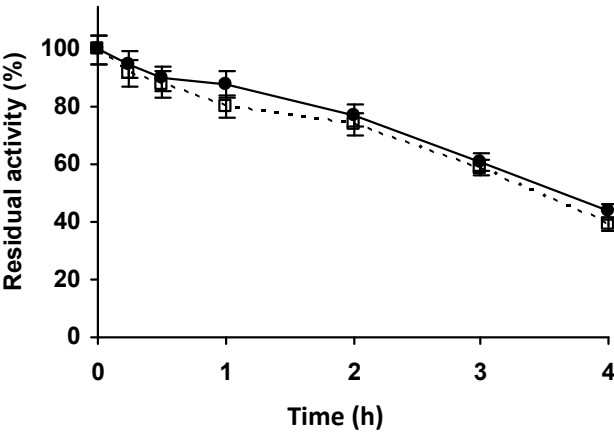

**Figure 9.** Thermal inactivation of r*Aae*UPO immobilized on MANAE agarose support. Inactivation was performed in 10 mM Tris at pH 7.0 and 57 °C. Other specifications can be found in the Methods section. Full circles: MANAE agarose biocatalyst; empty squares with dotted line: free enzyme.

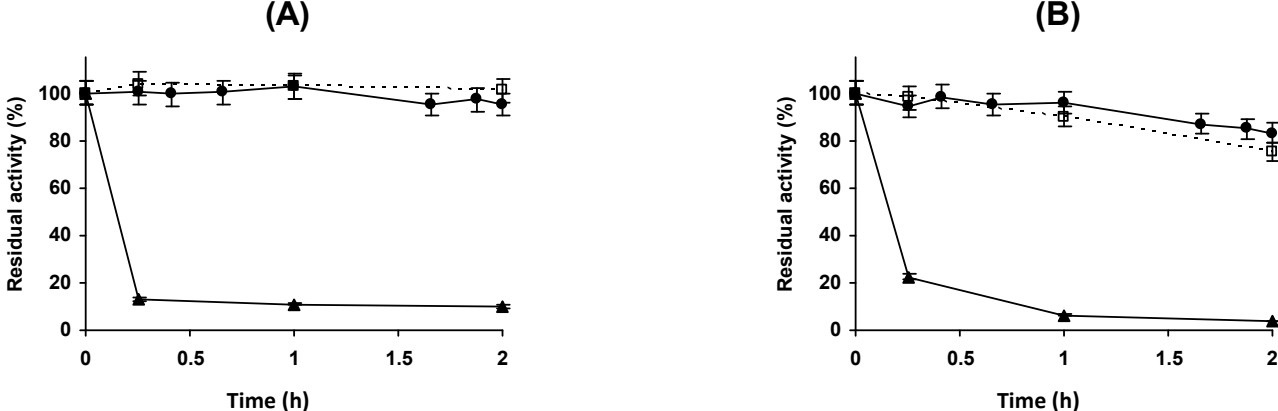

**Figure 10.** Immobilization of r*Aae*UPO on MANAE support at 25 °C and (**A**) pH 5.0 or (**B**) pH 9.0. Other specifications can be found in the Methods section. Full circles: suspension; full triangles: supernatant; empty squares with dotted line: reference.

The thermal inactivation courses of the 3 biocatalysts of the enzyme immobilized in MANAE agarose beads are shown in Figure 11. The enzyme immobilized at pH 5.0 and 7.0 gave very similar inactivation profiles to that of the free enzyme, while the enzyme immobilized at pH 9.0 was much less stable. This drastic difference of enzyme stability of enzymes immobilized in the same ion exchanger, but at different pH values, could be related to two factors: a different orientation of the enzyme in the support (involving different areas of the protein in the absorption when the immobilization pH changes and the ionization of the enzyme groups is altered) or the immobilization of different enzyme forms induced by the pH value [48,61,102]. The immobilization of lipases via interfacial activation on the same support (also a reversible immobilization protocol), but under different conditions, has proved to give different enzyme conformations, which are maintained after the immobilization [104–106].

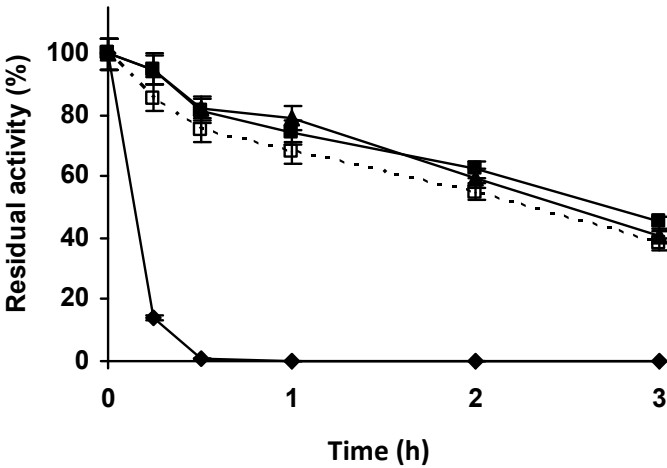

**Figure 11.** Thermal inactivation of r*Aae*UPO biocatalysts immobilized on MANAE agarose support at different pH. The experiment was performed in 10 mM Tris at 57 °C and pH 7.0. Other specifications can be found in the Methods section. Full triangles: MANAE biocatalyst prepared at pH 5.0; full squares: MANAE biocatalyst prepared at pH 7.0; full rhombus: MANAE biocatalyst prepared at pH 9.0; empty squares with dotted line: free enzyme.

In any case, this effect of the immobilization pH, even when the enzyme is reversibly immobilized, shows the significance of this parameter when preparing immobilized biocatalysts with this immobilization strategy.

Finally, we tried to improve the immobilized enzyme stability by treating the biocatalysts with glutaraldehyde, with the objective to establish enzyme–support crosslinking and, perhaps, some enzyme intermolecular crosslinking [39,48,50,102].

First, we treated the free enzyme with this reagent at pH 7.0 and 25 °C, (Figure 12A). The effect of this treatment on enzyme activity was not very significant using 0.1% or 1% glutaraldehyde; enzyme activity was maintained similarly to the activity of the enzyme incubated in absence of glutaraldehyde after 1 h (sufficient time to modify the primary amino groups with one glutaraldehyde molecule). When this treatment was performed on the enzymes adsorbed on MANAE support, the enzyme immobilized at pH 7.0 was fully inactivated after only 40 min, while the biocatalysts prepared by immobilizing the enzyme at pH 5.0 and 9.0 maintained more than 40% of the initial activity (Figure 12B). This also suggested the very different features of r*Aae*UPO ionically exchanged at different pH values. However, 24 h after washing the glutaraldehyde-modified biocatalysts, these biocatalysts were also fully inactive. These much more drastic effects of the glutaraldehyde on the activity of the enzyme adsorbed on aminated supports than on the free enzyme under similar treatment were fairly similar to those found using a similar strategy to immobilize ficin [49].

Thus, from the different protocols assayed in this paper, the optimal protocol for the immobilization of this enzyme is the immobilization of r*Aae*UPO on MANAE support at pH 5.0.

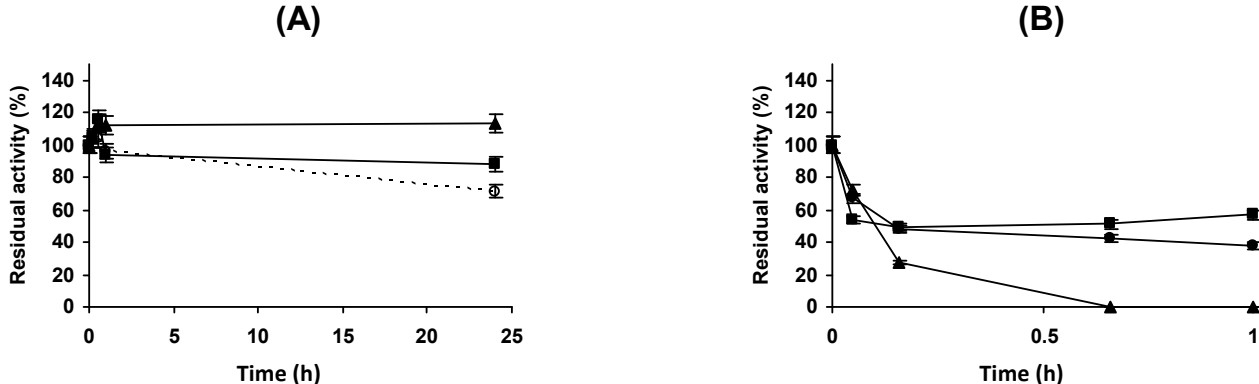

**Figure 12.** Effect of the incubation in presence of glutaraldehyde on the stability of different r*Aae*UPO preparations. (**A**) Free r*Aae*UPO. Full triangles: 0.1% glutaraldehyde; full squares: 1% glutaraldehyde; empty squares with dotted line: enzyme incubated without glutaraldehyde. (**B**) r*Aae*UPO-MANAE biocatalysts immobilized at different pH, incubated in presence of 0.1% glutaraldehyde. Full circles: MANAE biocatalyst prepared at pH 5.0; Full triangles: MANAE biocatalyst prepared at pH 7.0; Full squares: MANAE biocatalyst prepared at pH 9.0. The experiments were performed in 10 mM sodium phosphate at 25 °C and pH 7.0. Other specifications can be found in the Methods section.

### 2.4. Release of rAaeUPO from the MANAE Support and Reuse of the Support

One of the advantages of the immobilization of enzymes via ion exchange is the possibility of reuse of the support by releasing the enzyme after its inactivation. Figure 13 shows how the enzyme could be easily released from the support at both pH 4.4 and 7.0. Full enzyme release was achieved using 200 mM ammonium sulfate at pH 4.4 or 500 mM at pH 7.0. The support could be reused to immobilize fresh enzyme samples by 10 successive cycles, without any alteration of the enzyme performance.

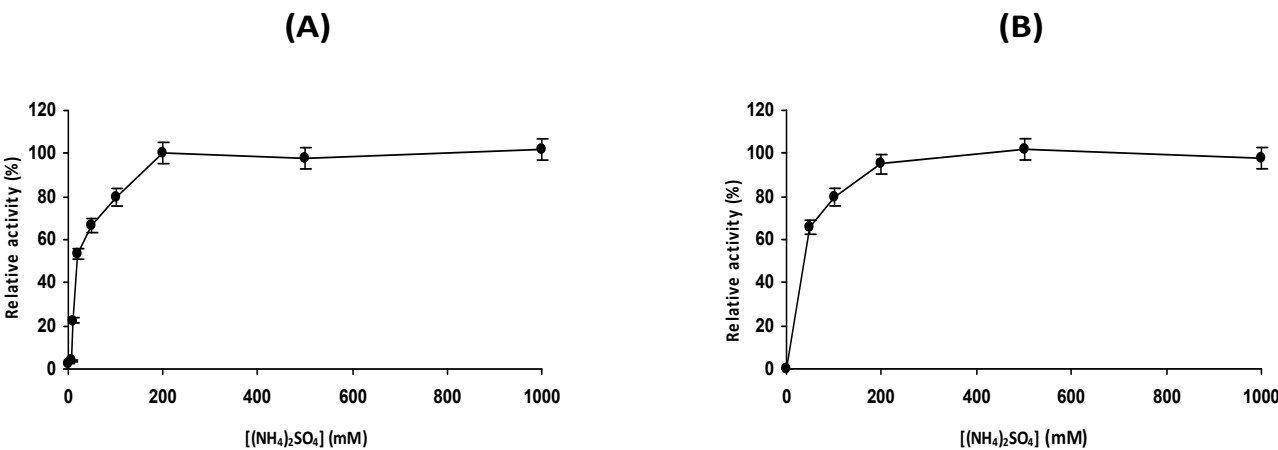

**Figure 13.** Desorption of *rAaeUPO* immobilized on MANAE support by incubation in growing ammonium sulfate concentration at pH 4.4 (**A**) and 7.0 (**B**). The experiments were performed as described in the Methods section.

### 2.5. Operational Stability of MANAE–rAaeUPO

r*Aae*UPO was immobilized on the MANAE support at pH 5.0, but in this instance using a loading of 3 mg/g. The immobilization yield was higher than 80% after 4 h. Using standard ABTS assay conditions, the enzyme was almost fully released from the support in the first cycle. Fifty mM citrate was enough to release most of the enzyme from the support (see Figure 13). Then, the experiment was repeated using 5 mM of sodium acetate at pH 5.0. Under these conditions, however, more than one third of the oxidized ABTS become adsorbed on the support and was released in the subsequent reuse, making reliable quantification of the new reaction course impossible. Washing steps using high ionic

strength buffers to remove the adsorbed ABTS oxidation product released the adsorbed oxidized ABTS, but it also resulted in enzyme leaching. A practical solution to study the operational stability of the biocatalyst was to reduce the concentration of ABTS to 0.03 mM. A reference of the previously used biocatalyst was incubated in the reaction medium (in the absence of fresh substrates) to determine the absorbance produced by the adsorbed during the previous cycles and now released oxidized ABTS. Then, this supernatant absorbance was subtracted from the absorbance of the new reaction cycle. We were therefore able to confirm that the immobilized enzyme could be reused in at least 10 reaction cycles without any significant reduction to activity (Figure 14). The activity in the 10th cycle was 75% of that of the initial one, very likely due to some enzyme release from the support even under these mild conditions.

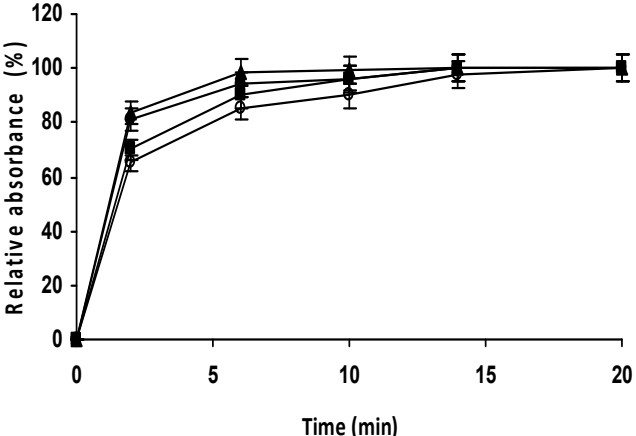

**Figure 14.** Operational stability of *rAaeUPO* immobilized on MANAE support. The reaction was performed for 20 min in 5 mM sodium acetate containing 0.03 mM ABTS and 0.125 mM hydrogen peroxide at pH 5.0, re-suspending the biocatalysts in a proportion of 1 g in 100 mL of buffer. The experiment was performed as described in the Methods section. The 100% absorbance was defined as the maximum absorbance achieved in the first cycle. Full circles: first reuse. Full triangles: third reuse. Full squares: sixth reuse. Empty circles: 10th reuse.

The experiment was performed to demonstrate that the reuse of the biocatalyst is possible, but it is clear that the oxidation of ABTS is not the target reaction for the utilization of this biocatalyst. It is evident that the biocatalyst will be of special interest for the application of the enzyme in anhydrous media, and there are many applications of this enzyme in these media, as discussed in the introduction. Its application in aqueous medium requires the use of very low concentrations of buffers and substrates.

## 3. Materials and Methods

### 3.1. Materials

Branched PEI (MW 10,000 and 25,000), sodium borohydride and glutaraldehyde solution Grade I, 25% in $H_2O$, were purchased from Sigma Aldrich (Alcobendas, Spain). BCL agarose bead standard (4%) was purchased from ABT (Burgos, Spain) and used to produce glyoxyl-agarose as described by Mateo et al. [37]. The amino-glutaraldehyde-agarose-activated support was prepared from monoaminoethyl-N-aminoethyl (MANAE) agarose [107] as previously described [50,108,109]. PEI coated agarose beads were prepared from glyoxyl supports as described by Mateo et al. [110]. The 2,2′-azino-bis(3-ethylbenzothiazoline-6-sulfonic acid) diammonium salt (ABTS®) was purchased from Roche (Mannheim, Germany). Divinyl sulfone (DVS) (97%) was purchased from Alfa Aesar (Heysham, UK). LMW-SDS (14.4–97.0 KDa) Marker for electrophoresis reference was purchased from GE Healthcare Life Sciences (Madrid, Spain). All other reagents were of analytical grade.

The recombinant and evolved unspecific peroxygenase from *Agrocybe aegerita* (r*Aae*UPO) was produced as described by Molina-Espeja et al. [64]. Figure S2 shows an SDS-PAGE of the relatively pure enzyme solution utilized in this research.

### 3.2. Methods

#### 3.2.1. Enzyme Preparation

The enzyme was produced via expression in *Pichia pastoris* as described by Molina-Espeja et al. [64]. The "unfiltered" solution contained *Pichia pastoris* cells and was centrifuged at 10,000 rpm at 4 °C for 20 min. The clarified supernatant was aliquoted and stored at −20 °C. The protein concentration ($8.85 \pm 0.15$ mg/mL) was determined through Bradford's method using bovine serum albumin as standard [111]. The semi-purified sample was utilized in all experiments. The enzyme features are similar to those described by Molina-Espeja et al. [64]; we did not re-characterize the enzyme sample.

#### 3.2.2. Determination of rAaeUPO Activity

The activity of r*Aae*UPO was determined via ABTS assay using a spectrophotometer thermoregulated at 25 °C with magnetic stirring. The ABTS assay was performed using as reaction medium 2.5 mL of 50 mM sodium citrate at pH 4.4 containing 2 mM $H_2O_2$ and 0.5 mM ABTS, adding the desired amount of r*Aae*UPO solution. The oxidation of ABTS was monitored by the change in absorbance at 418 nm ($\varepsilon_{418} = 36{,}000$ $M^{-1}$ $cm^{-1}$ under these conditions [81]). In this procedure, 25 µL of a r*Aae*UPO sample diluted in 50 mM Tris buffer at pH 7.0 was added to a cuvette containing 2 mL of the reaction solution, obtaining an activity of 1111 U per mL in the crude extract. One unit (U) of activity was defined as the amount of enzyme that oxidizes 1 µmol of substrate per minute under the specified conditions.

#### 3.2.3. SDS-PAGE Analysis

The SDS-PAGE analysis was carried out using 12% polyacrylamide gel with 5% polyacrylamide as concentrating gel [112]. The r*Aae*UPO samples were suspended in the rupture buffer (4% (*w/v*) SDS and 5% (*v/v*) mercaptoethanol in 200 mM Tris buffer at pH 7.0) to obtain final protein concentrations of 0.1 and 0.2 mg/mL. Subsequently, 15 µL of the samples were loaded on the gels, using 8 µL of low molecular weight marker protein as standard. After the separation, the gels were stained with Coomassie brilliant blue R-250, 3 mM in 40% (*v/v*) ethanol, and 10% (*v/v*) acetic acid for 1 h.

#### 3.2.4. Handling of the Enzyme

The enzyme solution was diluted in different buffers, at pH 7.0 (50 mM Tris, 50 mM HEPES or 50 mM sodium phosphate) and 10.0 (50 mM sodium carbonate) to a final concentration of 0.5 U/mL, using different temperatures. Periodically, samples were withdrawn and their activity assayed using ABTS as described above. In some instances, 20 or 40% (*v/v*) glycerol was added to the enzyme solutions.

#### 3.2.5. Immobilization of rAaeUPO

All experiments were performed at least in triplicate, and the results are presented as the mean values. Immobilization was characterized by immobilization course, immobilization yield and expressed activity characterized as recommended by Boudrant et al. [113]. Immobilization yield is defined as the percentage of the offered enzyme that is immobilized on the support (determined by the enzyme remaining in the supernatant and the activity maintained by the reference). Expressed activity is the percentage of observed activity of the immobilized enzyme considering the expected activity of the biocatalyst from the immobilization yield as 100%.

Immobilization of rAaeUPO on Glyoxyl-agarose Beads

Immobilization of r*Aae*UPO on this support was performed diluting r*Aae*UPO solution in 50 mM of sodium carbonate at pH 10.05 and 25 °C [40] with 20% or 40% glycerol to prevent the enzyme inactivation. The enzymatic activity of the suspension and supernatant was followed during the whole process [113].

Immobilization of rAaeUPO Via Anionic Exchange.

r*Aae*UPO solution was diluted in 10 mM sodium acetate at pH 5.0 (activity was 0.5 U/mL), 10 mM Tris at pH 7.0 or 10 mM sodium carbonate at pH 9.0 at 25 °C. Then, 1 g of MANAE- or PEI-activated agarose beads per 10 mL of enzyme solutions was added. It was kept in orbital stirring for 2 h. The enzymatic activity of the suspension and supernatant was followed during the whole process [113]. After the immobilization, the derivative was washed several times with distilled water and stored at 4–6 °C.

Immobilization of rAaeUPO on Glutaraldehyde-amino-agarose Support

Immobilization on glutaraldehyde-amino-agarose was performed using 1 g of support per 10 mL of enzyme solution (0.5 U/mL) prepared in 10 mM sodium acetate at pH 5.0, 10 mM sodium phosphate at pH 7.0 or 10 mM of sodium carbonate at pH 9.0 at 25 °C. The enzymatic activity of the suspension and supernatant was followed during the whole process using the ABTS assay described above [113]. Finally, the biocatalyst was washed with distilled water, vacuum dried and stored at 4–6 °C.

Immobilization on Vinyl Sulfone-agarose (VS-agarose) Support

The 4% BCL agarose beads were activated with DVS according to the methodology previously described by Santos et al. [114] with some modifications. Ten grams of agarose was added to 200 mL of a 333 mM sodium carbonate solution at pH 11.5 containing 350 mM of DVS. The resulting solution was kept under constant stirring for 2 h at 25 °C and finally was washed with excess distilled water and stored at 4–6 °C. r*Aae*UPO was immobilized using 50 mM sodium acetate at pH 5.0, 50 mM sodium phosphate at pH 7.0 or 50 mM sodium carbonate at pH 10.0 and 25 °C. During the immobilization, the activity of the supernatant, suspension and a reference under identical conditions was periodically determined [113].

3.2.6. Thermal Inactivation of the Different Biocatalysts

The free or immobilized enzyme samples were diluted/suspended in different buffers, pH and temperatures. Samples were periodically withdrawn, and the residual activity was quantified using the ABTS assay described above. The temperatures were selected to ensure reliable inactivation courses, and the error in the established temperatures was $\pm 1$ °C for different experiments using different water baths.

3.2.7. Desorption Assay of rAaeUPO from MANAE Support

r*Aae*UPO immobilized on MANAE agarose support as described above was suspended in 10 mM sodium phosphate at pH 7.0 or 10 mM sodium acetate pH 4.4, Then, solid ammonium sulphate was used to reach different concentrations. After 1 h of incubation, the activity of the suspension and the supernatant was measured to determine the percentage of released enzyme. A reference with free enzyme was subjected to the same treatment to determine any effect of the ammonium sulfate on enzyme activity (we found none).

3.2.8. Operational Stability of MANAE–rAaeUPO Biocatalyst

Lastly, 1 g of biocatalyst was suspended in 100 mL of 5 mM sodium acetate at pH 5.0 containing 0.125 mM of $H_2O_2$ and 0.03 mM of ABTS. To determine the absorbance, 2 mL of the reaction suspension was taken, and the biocatalyst was discarded by centrifugation at

6000 rpm for 2 min using an Eppendorf centrifuge. When the absorbance did not increase after 5 min, the biocatalysts were recovered by filtration and utilized in a new reaction cycle.

## 4. Conclusions

The immobilization of r*Aae*UPO is a quite complex task. This dimeric and highly glycosylated enzyme is destabilized by phosphate anions and stabilized by glycerol, and may be considered a quite stable enzyme (e.g., activity is fully maintained at pH 7.0 and 47 °C for 2 h). As it is a highly glycosylated enzyme, multipoint-covalent immobilization is a challenge. Production of the enzyme in prokaryotic expression systems resulting in non-glycosylated variants may offer the opportunity to further explore multi-point immobilization. This can over-compensate activity and stability losses stemming from the missing glycosylation pattern. The covalent immobilization strategies that we assayed failed (no immobilization or enzyme inactivation was observed) or produced very unstable biocatalysts. The ionic exchange of r*Aae*UPO in a polymeric ionic bed also produced very unstable immobilized enzyme biocatalysts. However, using MANAE agarose, immobilization yield was very high, and the enzyme maintained a high expressed activity. The immobilization pH slightly alters these parameters, but they are particularly relevant to the enzyme stability: The enzyme immobilized at pH 9.0 was less stable than the free enzyme. Thus, this immobilization condition, which was previously studied for its effect on enzyme activity or immobilization yield, should be considered a key condition in the determination of the immobilized enzyme features. However, even the best biocatalysts prepared in this research could only be used to maintain the activity and stability of the free enzyme, having as their main advantage the possibility of enzyme reuse. Other immobilization strategies, such as encapsulation (using sol-gel or MOFs) [115–118], crosslinked enzyme aggregates [119–121] or nanoflowers [122–124] could be explored with the objective of improving the current results.

**Supplementary Materials:** The following are available online at https://www.mdpi.com/article/10.3390/catal11050560/s1, **Figure S1**: Structure of rAaeUPO. The figure shows a ribbon structure of the dimers of r*Aae*UPO showing the Heme group in the active center (Red: Protoporphyrin IX containing Fe). **Figure S2**: SDS-PAGE in 12% polyacrylamide gel of r*Aae*UPO. The crude extract was centrifuged, aliquoted and stored at -20 °C. Samples were taken and diluted to the indicated concentrations. Other specifications can be found in Methods section. First lane: low molecular weight marker; Second lane: r*Aae*UPO at 2 mg/mL; Third lane: r*Aae*UPO at 1 mg/mL. **Figure S3**: Immobilization of r*Aae*UPO on glyoxyl-agarose support. The immobilizations were performed in 50 mM sodium carbonate at pH 10.05 and 25 °C in (A) absence of glycerol, (B) presence of 20% glycerol and (C) presence of 40% glycerol. Other specifications can be found in Methods section. Full circles: suspension; Full triangles: supernatant; Empty squares with dotted line: reference.

**Author Contributions:** Conceptualization, F.H. and R.F.-L.; methodology, R.F.-L.; investigation, D.C., R.M.-S., X.X.; writing—original draft preparation, D.C., R.M.-S., R.F.-L.; writing—review and editing, all authors; supervision, F.H., R.F.-L. All authors have read and agreed to the published version of the manuscript.

**Funding:** This research was funded by Ministerio de Ciencia e Innovación-Spanish Government (project number CTQ2017-86170-R), ERC consolidator grant, (No. 648026) and the Guangzhou Elite Project.

**Data Availability Statement:** All data needed to support the conclusions in the paper are contained in the article.

**Acknowledgments:** R.M.S. gives thanks to Ministerio de Educación -Spanish Government for an FPU fellowship and D.C. to Ministerio de Ciencia e Innovacion-Spanish Government for an FPI fellowship. X.X. and F.H. gratefully acknowledge financial support from the European Research Commission.

**Conflicts of Interest:** The authors declare no conflict of interest.

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
