# Peer review of "Immobilization of the Peroxygenase from Agrocybe aegerita. The Effect of the Immobilization pH on the Features of an Ionically Exchanged Dimeric Peroxygenase"

_catalysts, doi:10.3390/catal11050560_

Round 1

Reviewer 1 Report

In the present paper, the authors analyses different strategies to immobilize the recombinant, evolved variant of the dimeric unspecific peroxygenase from Agrocybe aegerita, an interesting catalyst for a broad of oxyfunctionalization reactions. The authors evaluated different methods for immobilizing the enzyme either covalently or by ionic interactions by monitoring the yield of immobilization and the stability of the immobilized enzyme, in particular with respect to temperature. Many interesting results seem to have been obtained, in particular original with regard to the influence of the immobilization pH on the efficiency of the process. However, the manuscript is difficult to read for lack of some details and important explanations.

Major points:

1- The authors use a recombinant enzyme produced in Pichia pastoris but no basic characterization is given: Several bands appear on the SDS-PAGE, is this expected? Is the heme cofactor present? Is the activity of the purified enzyme consistent with that described in the 2014 reference paper?

2- The authors took care to study the stability of the soluble form with respect to pH, temperature and buffer, but the results are not sufficiently convincing: the enzyme is said to be very stable at pH 7, 47 ° C for 35 hours without sources or results cited (line 162). Phosphate is proposed as destabilizing compared to Tris in view of the results presented in figure 2, but the results are not so clear since results in figure 2B and 3B are different yet it is supposed to be the same thing, the difference between phosphate and Tris is not obvious at 3 hours between figure 3A and 3B, so is the difference presented in figure 2 really significant? Finally, the stability of the enzyme is tested at 50 and 57°C but we don’t know why the authors have choice these values? In addition, why is it state “after 2 hours at 55°C and pH 7.0” in the abstract and conclusion sections? Same question for the choice of pH values (7 and 10 and not 5)? And for buffer (phosphate and Tris/Hepes and not carbonate) ?

3- Regarding immobilizations, the results are quite difficult to understand mainly because the authors interpret the "expressed activity" and the "immobilization yield" while these data do not appear in the figures and their method of calculation is not described. For example, page 6, “small immobilized fraction decreased its expressed activity by more than 50%” (figure 5A and 5B)?? At pH 9, “inactivation partially reduced (expressed activity was 40 %)”?? And results after 24 h not shown (I suppose?). Idem for explanation of fig 6. Finally, the results presented in figure 10 may be overinterpreted and interpretations should be moderate : expressed activity of 95 and 80% for immobilization at pH 5 and 9, respectively; immobilization yield of 90 and 100% for immobilization at pH 5 and 9, respectively. What is the threshold for detecting activity? What is the experimental error on these %? Are these differences significant?

4- The authors consider that immobilization by ionic interactions is reversible, it’s true in theory but in the case of their enzyme they have not test this. The authors also never discuss the effect of pH on enzyme activity in solution. These could be informative especially in term of explanation of the results regarding the potential existence of “different forms”. The last point concerns the result; the authors have succeeded in immobilizing this enzyme but apparently without gain in stability compared to the soluble form? Is there a difference in enzymatic properties?

Minor points:

  • The numbering of the last 14 references is false (page 17)
  • Figure 1 is nice but it does not in any way explain the absence of immobilization and it must be removed (like the text on page 6 lines 213 to 217) is illegible and not essential
  • The term MANAE has to be define.
  • The introduction section is too long.

Author Response

In the present paper, the authors analyses different strategies to immobilize the recombinant, evolved variant of the dimeric unspecific peroxygenase from Agrocybe aegerita, an interesting catalyst for a broad of oxyfunctionalization reactions. The authors evaluated different methods for immobilizing the enzyme either covalently or by ionic interactions by monitoring the yield of immobilization and the stability of the immobilized enzyme, in particular with respect to temperature. Many interesting results seem to have been obtained, in particular original with regard to the influence of the immobilization pH on the efficiency of the process. However, the manuscript is difficult to read for lack of some details and important explanations.

Major points:

The authors use a recombinant enzyme produced in Pichia pastoris but no basic characterization is given: Several bands appear on the SDS-PAGE, is this expected? Is the heme cofactor present? Is the activity of the purified enzyme consistent with that described in the 2014 reference paper?

A semi-purified sample of the enzyme has been utilized, as this has been produced as stated in the indicated reference, we have not re-characterized the enzyme. Heme cofactor should be present, as otherwise the enzyme should be inactive. We have indicated this in methods section..

2- The authors took care to study the stability of the soluble form with respect to pH, temperature and buffer, but the results are not sufficiently convincing: the enzyme is said to be very stable at pH 7, 47 ° C for 35 hours without sources or results cited (line 162).

This experiment was performed in our research, but we did not shown the figure as can be easily described in the test. We have now indicated that the results are not shown. We apologize by the confusing presentation.

Phosphate is proposed as destabilizing compared to Tris in view of the results presented in figure 2, but the results are not so clear since results in figure 2B and 3B are different yet it is supposed to be the same thing, the difference between phosphate and Tris is not obvious at 3 hours between figure 3A and 3B, so is the difference presented in figure 2 really significant?

Experiments in figure 3 are based in induvial experiments, are the referee is right, the Temperature in Tris was 57.5ºC while in phosphate was 57ºC, we have changed the figure legend..The differences in actual temperatures when using different baths is around 1ºC,  As this was a comparison of the effect of the dilution, we do not wanted to make a too complex legend, now we have clarified that. Also, we have leave clear the error of temperature in methods section.

Finally, the stability of the enzyme is tested at 50 and 57°C but we don’t know why the authors have choice these values? In addition, why is it state “after 2 hours at 55°C and pH 7.0” in the abstract and conclusion sections? Same question for the choice of pH values (7 and 10 and not 5)? And for buffer (phosphate and Tris/Hepes and not carbonate) ?

We have tried to explain that in the paper:

Temperatures have been chosen to have reliable inactivation courses, and our precision is ±1ºC.

pH were selected as relevant for enzyme immobilization and stabilization,  but we have added some information of the inactivation at pH 5.

The 55ºC was an additional experiment, in HEPES, but we agree that it is confusing, we have change to 35 h at 47ºC..

3- Regarding immobilizations, the results are quite difficult to understand mainly because the authors interpret the "expressed activity" and the "immobilization yield" while these data do not appear in the figures and their method of calculation is not described. For example, page 6, “small immobilized fraction decreased its expressed activity by more than 50%” (figure 5A and 5B)?? At pH 9, “inactivation partially reduced (expressed activity was 40 %)”?? And results after 24 h not shown (I suppose?). Idem for explanation of fig 6. Finally, the results presented in figure 10 may be overinterpreted and interpretations should be moderate : expressed activity of 95 and 80% for immobilization at pH 5 and 9, respectively; immobilization yield of 90 and 100% for immobilization at pH 5 and 9, respectively. What is the threshold for detecting activity? What is the experimental error on these %? Are these differences significant?

Sorry, we had supplied a reference for the terminology, but we agree that the terms need to be explained in the paper. This information has been added in the paper.

4- The authors consider that immobilization by ionic interactions is reversible, it’s true in theory but in the case of their enzyme they have not test this. The authors also never discuss the effect of pH on enzyme activity in solution. These could be informative especially in term of explanation of the results regarding the potential existence of “different forms”. The last point concerns the result; the authors have succeeded in immobilizing this enzyme but apparently without gain in stability compared to the soluble form? Is there a difference in enzymatic properties?

We have added the information on enzyme release from the support at two different pH, all enzyme could be easily released at both pH values. The studies of the activity versus pH using ABTS as substrate are not really interesting, but we have determined that at pH 7 the activity is only 10% of the activity at pH 4.4. We have added this information in the text. Our objective is not characterize this enzyme, just the features that are necessary to be able to handle and immobilize the enzyme, that have been expressed a relatively long time ago. A full characterization of this enzyme will be out of the scope of this paper, could be a biochemical independently paper.

Minor points:

  • The numbering of the last 14 references is false (page 17)

We apologize, one reference with number but without content was introduced in the final editing by error, this have been solved.

  • Figure 1 is nice but it does not in any way explain the absence of immobilization and it must be removed (like the text on page 6 lines 213 to 217) is illegible and not essential

This figure show that the lack of immobilization is not caused by a lack of reactive groups, so other explanation should be formulated. We will prefer to maintain this figure if the referee has not real problems, We have tried to improve the text.

  • The term MANAE has to be define.

It has been defined, sorry for this mistake.

  • The introduction section is too long.

Considering the number of immobilization strategies employed in this paper, we do not feel that to reduce the introduction is a simple task, as the information on all immobilization strategies employed should be supplied to leave clear that the failure in the immobilization is not because the supports are not suitable for enzyme immobilization.

Reviewer 2 Report

The manuscript of Carballares et al „Preparation of immobilized biocatalysts of a problematic enzyme. The effect of the immobilization pH on the features of an ionically exchanged dimeric peroxygenase” describes a study on immobilization of the recombinant dimeric unspecific peroxygenase from Agrocybe aegerita (rAaeUPO) on various supports including aminated agarose. Immobilization on the later support proved to be strongly pH dependent indicating the importance of pH upon immobilization. The enzyme immobilized at pH 7.0 or pH 5.0 presented good activity and stability very similar to that of the free enzyme, while immobilization at pH 9 resulted in almost inactive enzyme which was significantly less stable.

Notes:

  1. A discussion is missing from the Introduction clarifying why UPOs are important. UPOs catalyze various peroxide-dependent oxygenations and achieve catalytic efficiencies far beyond those of P450s and CPO. Please add a short section to this topic with proper examples and references.
  2. Introduction (line 77-82): for covalent multipoint attachment several functional groups are mentioned which can react with groups located in the enzyme surface (usually primary amino groups). Among those, glyoxyl, vinyl sulfone and glutaraldehyde activated supports are mentioned. I suggest extending this list with reference to epoxy-activated supports which are fine tunable and quite versatile and general (see also note 3).
  3. It is also a question why only the methods mentioned in note 1 and the ionic adsorption strategy were tried. In many cases, entrapment within a proper porous polymeric matrix proved to be successful. The entrapment strategy may also be helpful for enhancing the stability of multimeric enzymes.
  4. The authors list only two studies on immobilization of AaeUPO (covalent immobilization on epoxide supports resulting in low activity [ref 69]; and metal affinity immobilization of a His-tagged variant retaining respectable activity up to 78% [ref 70]). However, there exist several immobilization methods reported for unspecific peroxygenases which are not mentioned by the authors. AaeUPO could be immobilized by encapsulation in PVA/PEG cryogel and by retention of the enzyme in hollow fiber modules.[1] Structure-guided covalent immobilization could be successfully performed with strongly evolved AaeUPO could using properly activated carriers through one-point attachment.[2] A very recent study–including F. Hollmann among the co-authors–showed that AaeUPO immobilized on an epoxy-butyl-functionalized carrier could be the best protected from plasma exposure.[3] Please complete the Introduction and Discussion parts with these studies and perform a more rigorous comparison of the selected way of immobilization to the existing other methods.

Minor notes

- Check and correct the “degree C” in lines 163, 164, 167 (and in many later instances)

- Line 163: 35 h

- Line 212: pH 10.0 (Fig.

- Although the English of this MS is generally satisfactory for publication, it might be improved at certain parts by a thorough proofreading with fresh eyes.

Although the MS is relatively well prepared, the presented mode of immobilization is certainly not the only and the best one. Please perform a major revision in which you extend the comparison to other methods and better emphasize the importance of the enzyme and why these methods were selected. Therefore, I feel that this study can be accepted for publication only after a major revision.

Cited references:

  1. Poraj-Kobielska, M.; Peter, S.; Leonhardt, S.; Ullrich, R.; Scheibner, K.; Hofrichter, M. Immobilization of unspecific peroxygenases (EC 1.11.2.1) in PVA/PEG gel and hollow fiber modules. Biochem. Eng. J. 2015, 98, 144–150.
  2. Molina-Espeja, P.; Santos-Moriano, P.; García-Ruiz, E.; Ballesteros, A.; Plou, F.; Alcalde, M. Structure-Guided Immobilization of an Evolved Unspecific Peroxygenase. Int. J. Mol. Sci. 2019, 20, 1627.
  3. Yayci, A.; Dirks, T.; Kogelheide, F.; Alcalde, M.; Hollmann, F.; Awakowicz, P.; Bandow, J.E. Protection strategies for biocatalytic proteins under plasma treatment. J. Phys. D. Appl. Phys. 2021, 54, 035204.

Author Response

enzyme. The effect of the immobilization pH on the features of an ionically exchanged dimeric peroxygenase” describes a study on immobilization of the recombinant dimeric unspecific peroxygenase from Agrocybe aegerita (rAaeUPO) on various supports including aminated agarose. Immobilization on the later support proved to be strongly pH dependent indicating the importance of pH upon immobilization. The enzyme immobilized at pH 7.0 or pH 5.0 presented good activity and stability very similar to that of the free enzyme, while immobilization at pH 9 resulted in almost inactive enzyme which was significantly less stable.

Notes:

  1. A discussion is missing from the Introduction clarifying why UPOs are important. UPOs catalyze various peroxide-dependent oxygenations and achieve catalytic efficiencies far beyond those of P450s and CPO. Please add a short section to this topic with proper examples and references.

This has been added in the new version of the manuscript.

  1. Introduction (line 77-82): for covalent multipoint attachment several functional groups are mentioned which can react with groups located in the enzyme surface (usually primary amino groups). Among those, glyoxyl, vinyl sulfone and glutaraldehyde activated supports are mentioned. I suggest extending this list with reference to epoxy-activated supports which are fine tunable and quite versatile and general (see also note 3).

We agree that epoxy supports are also good alternative to have multipoint covalent immobilization, but we have not included them because they are not used in this paper (after the moderately bad results obtained in previous trials with this enzyme using this kind of supports). Considering that referee 1 asks for a reduction of the introduction, we think that add a support that is not later utilized may be unnecessary. However, if the referee considers this really relevant, we have not problems in adding information on these supportst.

  1. It is also a question why only the methods mentioned in note 1 and the ionic adsorption strategy were tried. In many cases, entrapment within a proper porous polymeric matrix proved to be successful. The entrapment strategy may also be helpful for enhancing the stability of multimeric enzymes.

Of course there are many different possibilities to immobilize an enzyme, and the encapsulation in sol-gels or MOFs, may be one valuable one. CLEA is another possibility, nanoflowers, etc. However, to use all available technologies   is out of any research, we have decided the use of preexisting supports in this first approach; other alternatives could be the subject of future researchers. We have added this possibility in conclusion section, including references.

  1. The authors list only two studies on immobilization of AaeUPO (covalent immobilization on epoxide supports resulting in low activity [ref 69]; and metal affinity immobilization of a His-tagged variant retaining respectable activity up to 78% [ref 70]). However, there exist several immobilization methods reported for unspecific peroxygenases which are not mentioned by the authors. AaeUPO could be immobilized by encapsulation in PVA/PEG cryogel and by retention of the enzyme in hollow fiber modules.[1] Structure-guided covalent immobilization could be successfully performed with strongly evolved AaeUPO could using properly activated carriers through one-point attachment.[2] A very recent study–including F. Hollmann among the co-authors–showed that AaeUPO immobilized on an epoxy-butyl-functionalized carrier could be the best protected from plasma exposure.[3] Please complete the Introduction and Discussion parts with these studies and perform a more rigorous comparison of the selected way of immobilization to the existing other methods.

Cited references:

  1. Poraj-Kobielska, M.; Peter, S.; Leonhardt, S.; Ullrich, R.; Scheibner, K.; Hofrichter, M. Immobilization of unspecific peroxygenases (EC 1.11.2.1) in PVA/PEG gel and hollow fiber modules. Biochem. Eng. J. 201598, 144–150.
  2. Molina-Espeja, P.; Santos-Moriano, P.; García-Ruiz, E.; Ballesteros, A.; Plou, F.; Alcalde, M. Structure-Guided Immobilization of an Evolved Unspecific Peroxygenase. Int. J. Mol. Sci. 201920, 1627.
  3. Yayci, A.; Dirks, T.; Kogelheide, F.; Alcalde, M.; Hollmann, F.; Awakowicz, P.; Bandow, J.E. Protection strategies for biocatalytic proteins under plasma treatment. J. Phys. D. Appl. Phys. 202154, 035204.

We have added the suggested references. Thank you for indicating them.

Minor notes

- Check and correct the “degree C” in lines 163, 164, 167 (and in many later instances)

Corrected

- Line 163: 35 h

Corrected

- Line 212: pH 10.0 (Fig.

pH is really 10.05 in these experiments

- Although the English of this MS is generally satisfactory for publication, it might be improved at certain parts by a thorough proofreading with fresh eyes.

 We have revised the manuscript.

Reviewer 3 Report

This manuscript descries the immobilization of rAaeUPO, a heme-containing peroxygenase. Based on the examination of several immobilization methods, the immobilization on PEI-support is effective. The finding will be a useful information. From the viewpoint of scientific contexts, the reviewer suggests the authors to rewrite the manuscript to improve the paper quality.

Introduction

Lines 83: ... the immobilization of unprotonated ...

Lines 136-146: Regarding the introduction of rAaeUPO, some essential information is missing: 

• Which heme cofactor is contained? (heme b? heme c? and so on)

• Figure 1: Where is the heme site (i.e. the reaction center)

• Figure 1: It is difficult to recognize that the enzymes has a dimer structure.

Results and discussion:

1) If the heme cofactor is heme b, the release of heme b is possible. If so, the binding stability of the heme must be investigated. 

2) If the heme center is located at the interface of protomers, the dissociation to monomers will be critical. The significance of the investigation on the enzyme concentration should be described connecting with the structural characteristics.

3) Although how to evaluate the activity is described in experimental section, the general chemical scheme should be shown in the text as a paper proposed to Catalysts

4) Does the immobilized enzyme readily release from the support? or stable?

Author Response

This manuscript descries the immobilization of rAaeUPO, a heme-containing peroxygenase. Based on the examination of several immobilization methods, the immobilization on PEI-support is effective. The finding will be a useful information. From the viewpoint of scientific contexts, the reviewer suggests the authors to rewrite the manuscript to improve the paper quality.

Introduction

Lines 83: ... the immobilization of unprotonated ...

Corrected

Lines 136-146: Regarding the introduction of rAaeUPO, some essential information is missing: 

  • Which heme cofactor is contained? (heme b? heme c? and so on)

It is heme B, we have included that in the paper.

  • Figure 1: Where is the heme site (i.e. the reaction center)

It is hidden, that way, it cannot be modified by the support during immobilization. We have remarked this in the paper.

  • Figure 1: It is difficult to recognize that the enzymes has a dimer structure.

We have added figure 1S to clearly show the dimeric nature of the enzyme, Figure 1 remains in the text to show the available groups.

Results and discussion:

1.- If the heme cofactor is heme b, the release of heme b is possible. If so, the binding stability of the heme must be investigated. 

This investigation, even although highly interesting to understand the mechanism of enzyme destabilization, is out of the scope of the paper, that is just about the immobilization of the enzyme. These experiments belong more to the biochemical characterization of the enzyme. However, we have not found any dependence on the enzyme stability with tits concentration, suggesting that dissociative processes are to the first step of the enzyme inactivation.

2) If the heme center is located at the interface of protomers, the dissociation to monomers will be critical. The significance of the investigation on the enzyme concentration should be described connecting with the structural characteristics.

Our results suggest that the dissociation of the enzyme subunits (or any other enzyme dissociation phenomena) is not the first step of the enzyme inactivation at pH 7, the promotion of a wrong assembly could be enough to explain the results, id facilitate the release of the prosthetic group.

3) Although how to evaluate the activity is described in experimental section, the general chemical scheme should be shown in the text as a paper proposed to Catalysts

ABTS assay is a very general peroxidase method. We have included a graphic with the general reaction scheme in introduction

4) Does the immobilized enzyme readily release from the support? or stable?

We have added this information (Figures 13 and 14)

Reviewer 4 Report

Submitted manuscript describes an original scientific work regarding develpoment of immobilization method for recombinant dimeric unspecific peroxygenase from Agrocybe aegerita and basic characterization of immobilized enzyme properties. It will be of interest for scientific community reading the journal Catalysts. I suggest to accept the manuscript for publication after minor revisions, which should by taken into account.

Formal issues:

Author should consider to simplify the title. The word „problematic“ is a colloquial rather than a technical term. What is problematic for one may not be for another. Similar is true for the word “good” in the term “good immobilization yield” in the Abstract (line 24), which should be omitted.

The meaning of the sentence „This decrease in enzyme activity could be explained by a certain enzyme-support multipoint covalent attachment, which could be interesting if it is correlated with an increase in enzyme stability [30].“ is not clear and should be rewritten. (lines 244-246).

„Molina-Espeja et al. [82].“ (line 381) change to „Molina-Espeja et al. [83].“

change „sugar“ to „saccharide“ in the whole text

Scientific remarks:

The text is descriptive rather than fluent. In general, the design of experiments gives the impression of trial and error method. However, obtained final immobilization method is usable. Therefore it is pity, that there is no description of potential utilisation of obtained biocatalyst including industrial production of certain class of compounds and so on. The latter should be added into discussion.

The reason to use certain temperature values 50 an 57°C (fig. 2) is missing. What was the reason to use such a narrow temperature difference?

There is almost no difference in the enzyme deactivation pattern in Figure 6. What was the reason to design the experiments only for pH 5 and pH 7, and not more values in wider range of pH?

One of the main characteristics for immobilized enzymes for production of valuable compounds is their operational stability during repeated biotransformation cycles. Why there was no such experiment?

It is known feature, the expression of enzymes in Pichia pastoris may produce glycosylation of recombinant enzyme (which is true also for S. cerevisiae). This glycosylation may have major impact on ability of enzyme to form multipoint attachment as described in the manuscript (line 102). Why Pichia pastoris was selected as an expression system for the peroxygenase? What is opinion of authors on using Escherichia coli as expression system? These phenomenons should be explained in the discussion.

Author Response

Submitted manuscript describes an original scientific work regarding develpoment of immobilization method for recombinant dimeric unspecific peroxygenase from Agrocybe aegerita and basic characterization of immobilized enzyme properties. It will be of interest for scientific community reading the journal Catalysts. I suggest to accept the manuscript for publication after minor revisions, which should by taken into account.

Formal issues:

Author should consider to simplify the title. The word „problematic“ is a colloquial rather than a technical term. What is problematic for one may not be for another. Similar is true for the word “good” in the term “good immobilization yield” in the Abstract (line 24), which should be omitted.

We have revised title and abstract as suggest

The meaning of the sentence „This decrease in enzyme activity could be explained by a certain enzyme-support multipoint covalent attachment, which could be interesting if it is correlated with an increase in enzyme stability [30].“ is not clear and should be rewritten. (lines 244-246).

 We have rewritten the sentence.

„Molina-Espeja et al. [82].“ (line 381) change to „Molina-Espeja et al. [83].“

References had  a problem since reference 78, it has been solved in this new version, We apologize by the mistake

change „sugar“ to „saccharide“ in the whole text

 This has been changed as suggested.

Scientific remarks:

The text is descriptive rather than fluent. In general, the design of experiments gives the impression of trial and error method. However, obtained final immobilization method is usable. Therefore it is pity, that there is no description of potential utilisation of obtained biocatalyst including industrial production of certain class of compounds and so on. The latter should be added into discussion.

 We have added some reuses experiments as commented by other referees, and also added some recommendations for the use of this biocatalyst. Future research intend to use this biocatalyst in anhydrous media.

The reason to use certain temperature values 50 an 57°C (fig. 2) is missing. What was the reason to use such a narrow temperature difference?

 Just conditions where a reliable and not too slow inactivation course was obtained, we have included that in the paper. Higher temperatures produce a too rapid inactivation; a lower temperature would give a too slow inactivation. We have explained this in methods.

There is almost no difference in the enzyme deactivation pattern in Figure 6. What was the reason to design the experiments only for pH 5 and pH 7, and not more values in wider range of pH?

The failure was due to enzyme inactivation, and the enzyme was stable at pH 5-7, the use of alkaline pH value could improve the intensity of the multipoint covalent attachment, but with the results at pH 5 and 7 it was evident that results should be worse at pH 9. In fact, we have preformed this experiment and results were, as expected even worse than at pH 7. We have included that now in the paper.

One of the main characteristics for immobilized enzymes for production of valuable compounds is their operational stability during repeated biotransformation cycles. Why there was no such experiment?

 We have added this experiment, although it was difficult due to the adsorption of oxidized ABTS on the support.

It is known feature, the expression of enzymes in Pichia pastoris may produce glycosylation of recombinant enzyme (which is true also for S. cerevisiae). This glycosylation may have major impact on ability of enzyme to form multipoint attachment as described in the manuscript (line 102). Why Pichia pastoris was selected as an expression system for the peroxygenase? What is opinion of authors on using Escherichia coli as expression system? These phenomenons should be explained in the discussion.

We find this very interesting; we have included some discussion on this topic. Thank you for the suggestion.

Round 2

Reviewer 3 Report

I confirmed that the authors had revised the manuscript according to my previous suggestions. I can recommend this manuscript for publication as is.

Author Response

Thank you!